# PeerNets: Exploiting Peer Wisdom Against Adversarial Attacks

**Jan Svoboda[1,2], Jonathan Masci[2], Federico Monti[1,4], Michael M. Bronstein[1,3,4,5], Leonidas Guibas[6]**
[1]USI, Switzerland     [2]NNAISENSE, Switzerland     [3]Intel Perceptual Computing, Israel
[4]Imperial College London, UK     [5]Fabula AI, UK     [6]Stanford University, USA
`{jan.svoboda,federico.monti,michael.bronstein}@usi.ch`
`jonathan@nnaisense.com, guibas@cs.stanford.edu`

## Abstract

Deep learning systems have become ubiquitous in many aspects of our lives. Unfortunately, it has been shown that such systems are vulnerable to adversarial attacks, making them prone to potential unlawful uses. Designing deep neural networks that are robust to adversarial attacks is a fundamental step in making such systems safer and deployable in a broader variety of applications (e.g. autonomous driving), but more importantly is a necessary step to design novel and more advanced architectures built on new computational paradigms rather than marginally building on the existing ones. In this paper we introduce **PeerNets**, a novel family of convolutional networks alternating classical Euclidean convolutions with graph convolutions to harness information from a graph of peer samples. This results in a form of non-local forward propagation in the model, where latent features are conditioned on the global structure induced by the graph, that is up to $3\times$ more robust to a variety of white- and black-box adversarial attacks compared to conventional architectures with almost no drop in accuracy.

## 1 Introduction

Deep convolutional networks (CNN) are the de-facto standard in almost any computer vision application, ranging from image recognition (Krizhevsky et al., 2012; He et al., 2015), object detection (Yuan et al., 2017; Redmon et al., 2016; Ren et al., 2015), semantic segmentation (He et al., 2017; Badrinarayanan et al., 2017) and motion estimation (Liu et al., 2017; Vijayanarasimhan et al., 2017; Niklaus et al., 2017). The ground breaking results of deep learning for industry-relevant problems have made them also an important ingredient in many real world systems for applications such as autonomous driving and user authentication. Unfortunately, it has been shown (Szegedy et al., 2013) that such systems are vulnerable to easy to fabricate adversarial examples opening potential ways for their untraceable and unlawful exploitation.

**Adversarial attacks.** Szegedy et al. (2013) found that deep neural networks employed in computer vision tasks tend to learn very discontinuous input-output mappings, and can be caused to misclassify an image by applying an almost imperceptible 'adversarial' perturbation, which is found by maximizing the network's prediction error. While it was expected that addition of noise to the input can ruin the classification accuracy of a neural network, only tiny adversarial perturbations are needed as a great surprise to the machine learning community. Multiple methods for designing adversarial perturbations have been proposed, perhaps the most dramatic being single pixel perturbation by Su et al. (2017). Because of its potentially dramatic implications on the security of deep learning technology (which are employed for example, in face recognition systems Sharif et al. (2016) or autonomously driving cars), adversarial attacks and defenses against them have become a very active topic of research (Rauber et al., 2017; Kurakin et al., 2018).

Adversarial attacks can be categorized as *targeted* and *non-targeted*. The former aim at changing a source data sample in a way to make the network classify it as some pre-specified target class (imagine as an illustration a villain that would like a computer vision system in a car to classify all traffic signs as STOP signs). The latter type of attacks, on the other hand, aims simply at making the classifier predict some different class. Furthermore, we can also distinguish between the ways in

which the attacks are generated: *white box* attacks assume the full knowledge of the model and of all its parameters and gradient, while in *black box* attacks, the adversary can observe only the output of a given generated sample and has no access to the model. It has been recently found that for a given model and a given dataset there exists a *universal adversarial perturbation* by Moosavi-Dezfooli et al. (2016) that is network-specific but input-agnostic, i.e., a single noise image that can be applied to any input of the network. Such perturbations were furthermore shown to be insensitive to geometric transformations.

**Defense against adversarial attacks.** There has been significant recent effort trying to understand the theoretical nature (Goodfellow et al., 2015; Fawzi et al., 2018) and strategies against adversarial perturbations. The search for solution has focused on the network architecture, training procedure, and data pre-processing (Gu & Rigazio, 2014). Intuitively, one could argue that robustness to adversarial noise could be achieved by adding adversarial examples during training. However, recent work showed that this brute-force approach does not work in this case and that it forces the network to converge to a bad local minimum (Tramèr et al., 2018). As an alternative, Tramèr et al. (2017) introduced ensemble adversarial training augmenting training data with perturbations transferred from other models. Papernot & McDaniel (2016) proposed defensive distillation training. Mahdizadehaghdam et al. (2018) showed that deep dictionary learning architecture provides greater robustness against adversarial noise. Lately, Athalye et al. (2018) have pointed to the problem of so called *gradient obfuscation*, which has to be carefully treated during design of a new defense method. They have proven it to be the main reason why many state-of-the-art methods work, which further emphasizes the fact that defense against adversarial attacks are still an open question.

**Main contributions.** In this paper, we introduce *Peer-Regularized Networks* (PeerNet), a new family of deep models that uses a graph of samples to perform *non-local forward propagation*. The output of the network depends not only by the given test sample at hand, but also by its interaction with several training samples. We experimentally show that such a novel paradigm leads to substantially higher robustness (up to $3\times$) to adversarial attacks at the cost of a minor classification accuracy drop when using the same architecture. We also show that the proposed non-local propagation acts as a strong regularizer that makes possible to increase the model capacity to match the current state-of-the-art without incurring in overfitting. We provide experimental validation on established benchmarks such as MNIST, CIFAR10 and CIFAR100 using various types of adversarial attacks.

## 2 RELATED WORKS

Our approach is related to three classes of approaches: deep learning on graphs, deep learning models combined with manifold regularization, and non-local filtering. In this section, we review the related literature and provide the necessary background.

### 2.1 DEEP LEARNING ON GRAPHS

In the recent years, there has been a surge of interest in generalizing successful deep learning models to non-Euclidean structured data such as graphs and manifolds, a field referred to as *geometric deep learning* in Bronstein et al. (2017). First attempts of learning on graphs date back to the works of Gori et al. (2005); Scarselli et al. (2009), where the authors considered steady state of learnable diffusion process (more recent works Li et al. (2016); Gilmer et al. (2017) improved this approach using modern deep learning schemes).

**Spectral domain graph CNNs.** Bruna et al. (2013); Henaff et al. (2015) proposed formulating convolution-like operations in the spectral domain, defined by the eigenvectors of the graph Laplacian. Among the drawbacks of this architecture is $\mathcal{O}(n^2)$ computational complexity due to the cost of computing the forward and inverse graph Fourier transform, $\mathcal{O}(n)$ parameters per layer, and no guarantee of spatial localization of the filters.

A more efficient class of spectral graph CNNs was introduced in Defferrard et al. (2016); Kipf & Welling (2016) and follow up works, who proposed spectral filters that can be expressed in terms of simple operations (such as additions, scalar- and matrix multiplications) w.r.t. the Laplacian. In particular, Defferrard et al. (2016) considered polynomial filters of degree $p$, which incur only $p$ times multiplication by the Laplacian matrix (which costs $\mathcal{O}(|\mathcal{E}|)$ in general or $\mathcal{O}(n)$ if the graph is sparsely connected), also guaranteeing filters that are supported in $p$-hop neighborhoods. Levie

et al. (2017) proposed rational filter functions including additional inversions of the Laplacian, which were carried our approximately using an iterative method. Monti et al. (2018) used multivariate polynomials w.r.t. multiple Laplacians defined by graph motifs; Monti et al. (2017b) used Laplacians defined on products of graphs in the context of matrix completion problems.

**Spatial domain graph CNNs.** On the other hand, spatial formulations of graph CNNs operate on local neighborhoods on the graph (Duvenaud et al., 2015; Monti et al., 2017a; Atwood & Towsley, 2016; Hamilton et al., 2017; Veličković et al., 2017; Wang et al., 2018). Monti et al. (2017a) proposed the Mixture Model networks (MoNet), generalizing the notion of image 'patches' to graphs. The centerpiece of this construction is a system of local pseudo-coordinates $\mathbf{u}_{ij} \in \mathbb{R}^d$ assigned to a neighbor $j$ of each vertex $i$. The spatial analogue of a convolution is then defined as a Gaussian mixture in these coordinates. Veličković et al. (2017) reinterpreted this scheme as *graph attention* (GAT), learning the relevance of neighbor vertices for the filter result,

$$\tilde{\mathbf{x}}_i = \text{ReLU}\left(\sum_{j \in \mathcal{N}_i} \alpha_{ij}\mathbf{x}_j\right), \qquad \alpha_{ij} = \frac{\exp(\text{LeakyReLU}(\mathbf{b}^\top[\mathbf{Ax}_i, \ \mathbf{Ax}_j]))}{\sum_{k \in \mathcal{N}_i}\exp(\text{LeakyReLU}(\mathbf{b}^\top[\mathbf{Ax}_i, \ \mathbf{Ax}_j]))} \tag{1}$$

where $\alpha_{ij}$ are attention scores representing the importance of vertex $j$ w.r.t. $i$, and the $p \times q'$ matrix $\mathbf{A}$ and $2p$-dimensional vector $\mathbf{b}$ are the learnable parameters.

## 2.2 Low-Dimensional Regularization

There have recently been several attempts to marry deep learning with classical manifold learning methods (Belkin & Niyogi, 2003; Coifman & Lafon, 2006) based on the assumption of local regularity of the data, that can be modeled as a low-dimensional manifold in the data space. Zhu et al. (2017) proposed a feature regularization method that makes input and output features to lie on a low dimensional manifold. Garcia & Bruna (2017) cast few shot learning as supervised message passing task which is trained end-to-end using graph neural networks. One of the key disadvantages of such approaches is the difficulty to factor out global transformations (such as translations) to which the network output should be invariant, but which ruin the local similarity.

## 2.3 Non-Local Image Filtering

The last class of methods related to our approach are non-local filters Sochen et al. (1998); Tomasi & Manduchi (1998); Buades et al. (2005) that gained popularity in the image processing community about two decades ago. Such non-shift-invariant filters produce, for every location of the image, a result that depends not only on the pixel intensities around the point, but also on a set of neighboring pixels and their relative locations. For example, the bilateral filter uses radiometric differences together with Euclidean distance of pixels to determine the averaging weights. The same paradigm has brought into Non-local networks by Wang et al. (2017), where the response at a given position is defined as a weighted sum of all the features across different spatio-temporal locations, whose interpolation coefficients are inferred by a parametric model, usually another neural network.

## 3 Peer Regularization

In this paper, we propose a new deep neural network architecture that takes advantage of the data space structure. The centerpiece of our model is the *Peer Regularization* (PR) layer, designed as follows. Let $\mathbf{X}^1, \ldots, \mathbf{X}^N$ be $n \times d$ matrices representing the feature maps of $N$ images, to which we refer as *peers* (here $n$ denotes the number of pixels and $d$ is the dimension of the feature in each pixel). Given a pixel of image $i$, we consider its $K$ nearest neighbor graph in the space of $d$-dimensional feature maps of all pixels of all the peer images, where the neighbors are computed using e.g. the cosine distance. The $k$th nearest neighbor of the $p$th pixel $\mathbf{x}_p^i$ taken from image $i$ is the $q_k$th pixel $\mathbf{x}_{q_k}^{j_k}$ taken from peer image $j_k$, with $k = 1, \ldots, K$ and $j_k \in \{1, \ldots, N\}$, $q_k \in \{1, \ldots, n\}$.

We apply a variant of graph attention network (GAT) by Veličković et al. (2017) to the nearest-neighbor graph constructed this way,

$$\tilde{\mathbf{x}}_p^i = \sum_{k=1}^K \alpha_{ij_kpq_k}\mathbf{x}_{q_k}^{j_k}, \qquad \alpha_{ij_kpq_k} = \frac{\text{LeakyReLU}(\exp(a(\mathbf{x}_p^i, \mathbf{x}_{q_k}^{j_k})))}{\sum_{k'=1}^K \text{LeakyReLU}(\exp(a(\mathbf{x}_p^i, \mathbf{x}_{q_{k'}}^{j_{k'}})))} \tag{2}$$

where $a()$ denotes a fully connected layer mapping from $2d$-dimensional input to scalar output, and $\alpha_{ij_k pq_k}$ are attention scores determining the importance of contribution of the $q_k$th pixel of image $j$ to the output $p$th pixel $\tilde{\mathbf{x}}_p^i$ of image $i$. This way, the output feature map $\tilde{\mathbf{X}}^i$ is pixel-wise weighted aggregate of the peers. Peer Regularization is reminiscent of non-local means denoising of Buades et al. (2005), with the important difference that the neighbors are taken from multiple images rather than from the same image, and the combination weights are learnable.

**Full graph approximation.** In principle, one would use a graph built out of a nearest neighbor search across all the available training samples. Unfortunately, this is not feasible due to memory and computation limitations. We therefore use a Monte Carlo approximation, as follows. Let $\mathbf{X}^1, \dots, \mathbf{X}^{N'}$ denote the images of the training set. We select randomly with uniform distribution $M$ smaller batches of $N \ll N'$ peers, batch $m$ containing images $\{l_{m1}, \dots, l_{mN}\} \subset \{1, \dots, N'\}$. The nearest-neighbor graph is constructed separately for each batch $m$, such that the $k$th nearest neighbor of pixel $p$ in image $i$ is pixel $q_{mk}$ in image $j_{mk}$, where $m = 1, \dots, M$, $j_{mk} \in \{l_{m1}, \dots, l_{mN}\}$, and $q_{mk} \in \{1, \dots, n\}$. The output of the filter is approximated by empirical expectation on the $M$ batches, estimation as follows

$$\tilde{\mathbf{x}}_p^i = \frac{1}{M} \sum_{m=1}^{M} \sum_{k=1}^{K} \alpha_{ij_{mk}pq_{mk}} \mathbf{x}_{q_{mk}}^{j_{mk}}. \tag{3}$$

In order to limit the computational overhead, $M = 1$ is used during training, whereas larger values of $M$ are used during inference (see Section 4 for details).

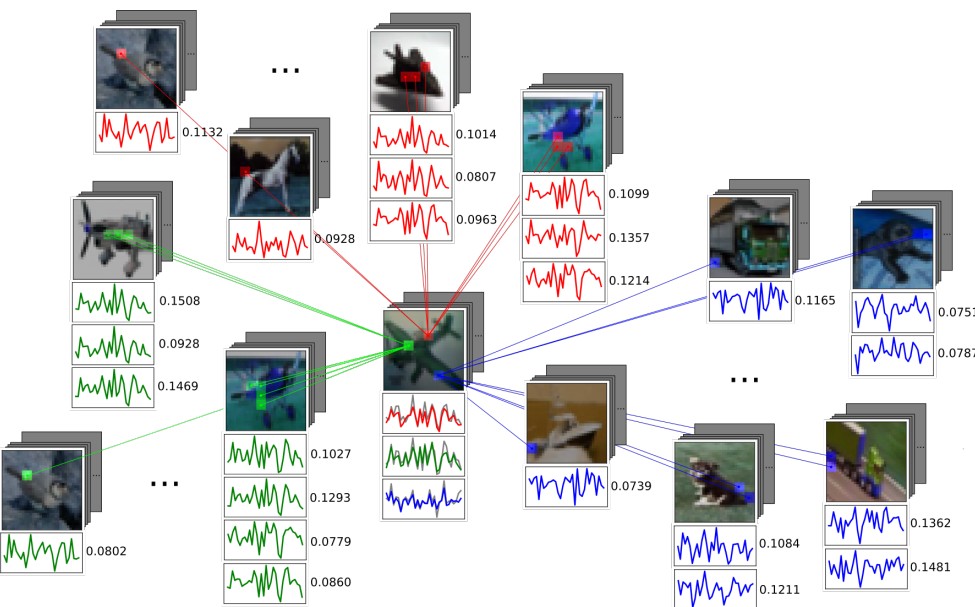

Figure 1: Our Peer Regularization illustrated on three pixels (red, green, blue) of a CIFAR image (center). For each pixel, five nearest neighbors are found in peer images. Plots represent the feature maps in the respective pixels; numbers represent the attention scores.

## 4 EXPERIMENTS

We evaluate the robustness of Peer Regularization to adversarial perturbations on standard benchmarks (MNIST LeCun (1998), CIFAR-10, and CIFAR-100 Krizhevsky (2009)) using a selection of common architectures (LeNet by LeCun et al. (1998) and ResNet by He et al. (2015)). The modifications of the aforementioned architectures with the additional PR layers are referred to as PR-LeNet and PR-ResNet and depicted in Figure 2. Additional details and results appear in the appendices.

### 4.1 GRAPH CONSTRUCTION

**Training.** During training, the graph is constructed using all the images of the current batch as peers. We compute all-pairs distances and select $K$-nearest neighbors. To mitigate the influence of the small batch sizes during training of PR-Nets, we introduce high level of stochasticity inside the PR layer, by using dropout of $0.2$ on the all-pairs distances while performing $K$-nearest neighbors and $0.5$ on the attention weights right before the *softmax* nonlinearity. We used $K = 10$ for all the experiments.

**Testing.** For testing, we select $N$ fixed peer images and then compute distances between each testing sample and all the $N$ peers. Feeding a batch of test samples, each test sample can be adjacent only to the $N$ samples in the fixed graph, and not to other test samples. Because of the approximation in the graph construction (see Section 3), we have to ensure that our random pre-selection of the graph nodes does not influence the performance. We therefore perform a Monte Carlo sampling over $M$ forward passes with different uniformly sampled graphs with $N$ nodes and we average over the $M$ runs. We used $N = 50$ for MNIST and CIFAR-10 and $N = 500$ for CIFAR-100, over $M = 10$ runs.

### 4.2 ARCHITECTURES

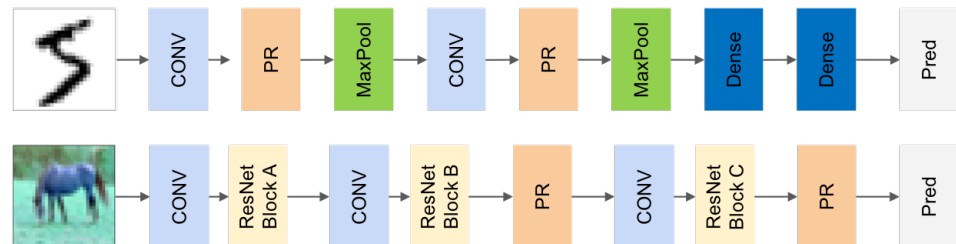

Figure 2: *Top:* PR-LeNet architecture used on the MNIST dataset; *Bottom:* PR-ResNet architecture used for the CIFAR experiments, letters A, B, and C indicate the number of feature maps in each residual block. The respective baseline models are produced by simply removing the PR layers.

**MNIST.** We modify the LeNet-5 architecture by adding two PR layers, after each convolutional layer and before max-pooling (Figure 2, top).

**CIFAR-10.** We modify the ResNet-32 model, with $A = 16$, $B = 32$ and $C = 64$, as depicted in Fig. 2 by adding two PR layers at the last change of dimensionality and before the classifier. Each ResNet block is a sequence of Conv + BatchNorm + ReLU + Conv + BatchNorm + ReLU layers.

**CIFAR-100.** For this dataset we take ResNet-110 ($A = 16$, $B = 32$ and $C = 64$) and modify it in the same way as for CIFAR-10. Each block is of size 18.

### 4.3 RESULTS

We tested robustness against several different types of white-box adversarial attacks: gradient descent, fast-gradient sign method of Goodfellow et al. (2015), projected gradient descent and universal adversarial perturbation of Moosavi-Dezfooli et al. (2015). Similarly to previous works on adversarial attacks, we evaluate robustness in terms of the *fooling rate*, defined as the ratio of images for which the network predicts a different label as the result of the perturbation. As suggested in Moosavi-Dezfooli et al. (2016), random perturbations aim to shift the data points to the decision boundaries of the classifiers. For the method of Moosavi-Dezfooli et al. (2016), even if the perturbations should be universal across different models, we generated it for each model separately to achieve the fairest comparison.

**Timing.** On CIFAR-10 our unoptimized PR-ResNet-32 implementation, during training, processes a batch of 64 samples in $0.15s$ compared to the ResNet-32 baseline that takes $0.07s$. At inference time a batch of 100 samples with a graph of size 100 is processed in $1.5s$ whereas the baseline takes $0.4s$. However, it should be noted that much can be done to speed-up the PR layer, we leave it for future work.

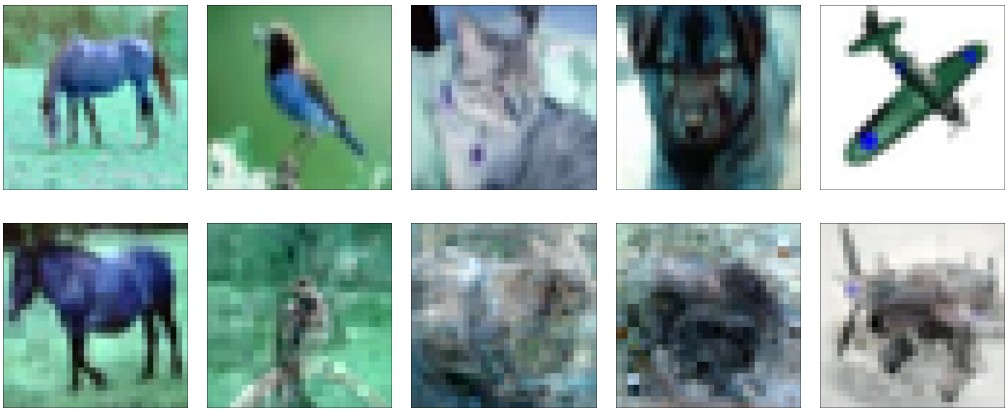

Figure 3: Examples of "Franken-images" (second row) constructed by backpropagating the attention scores to the input and using them to compose a new image as weighted sum of the peer pixels. Original images are shown in the first row.

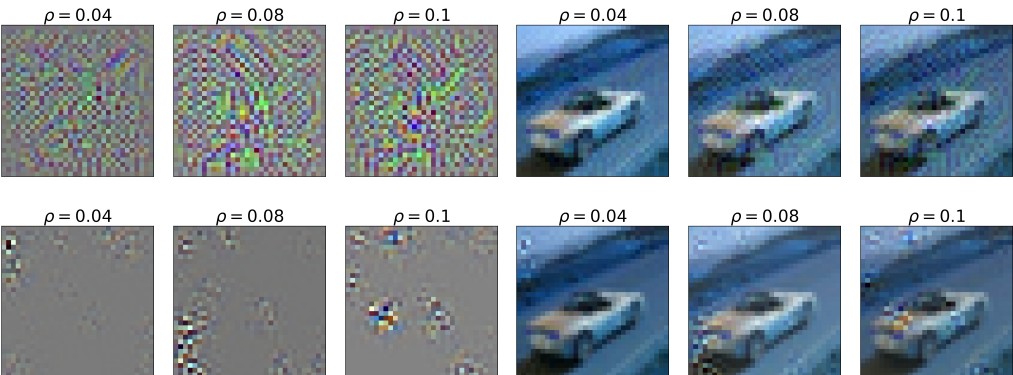

Figure 4: Examples of generated universal adversarial perturbations for CIFAR-10 dataset (left) and their applications to a sample image (right) for different values of $\rho$. Shown are results of ResNet-32 (top) and PR-ResNet-32 (bottom).

**Gradient obfuscation.** With regard to recent work of Athalye et al. (2018), we make sure that PeerNet models yield reasonable gradients during perturbation generation, and therefore do not suffer from this phenomenon. Following experiments further support our claim, as they show that PeerNets do not exhibit the behaviour that typically accompanies this problem (e.g. our model can be always fooled with an unbounded attack, iterative attacks are more efficient than the single-step ones, etc. ).

### 4.3.1 SAMPLE-SPECIFIC ATTACKS

Let us first evaluate our model on sample-specific non-targeted attacks. This means, having a testing sample with label $y$, we aim to generate an adversarial sample particularly for this testing sample, which will have label $\hat{y} \neq y$, while satisfying $||\mathbf{x} - \hat{\mathbf{x}}||_\infty \leq \epsilon$, where $\epsilon$ is some small value to produce a perturbation $\mathbf{v}$ that is nearly perceivable. For best reproducibility we used implementations of the following methods provided in Foolbox by Rauber et al. (2017).

**Gradient Descent attack.** Generation of the adversarial example is posed as a constrained optimization problem and solved using gradient descent. It can be summarized by the following formula:

$$\hat{\mathbf{x}} \leftarrow \hat{\mathbf{x}} + \epsilon \cdot \nabla \log p(\hat{y}|\hat{\mathbf{x}})$$

where ten $\epsilon$ values are scanned in ascending order, until the algorithm either succeeds or fails.

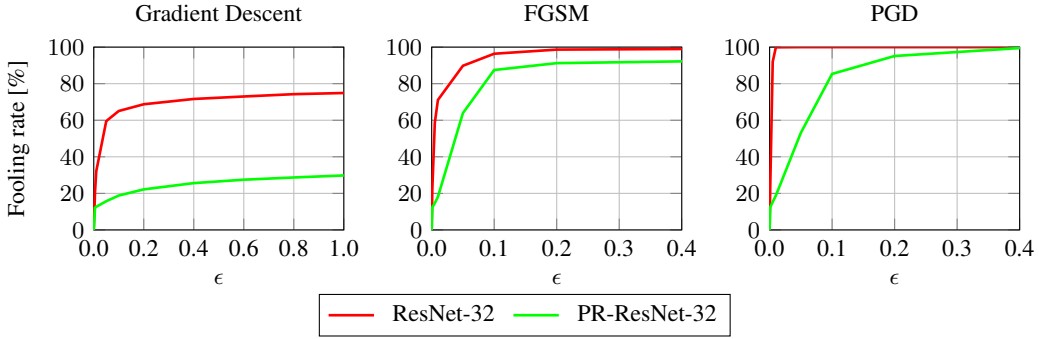

Figure 5: Comparison of PeerNet and classical CNN on the CIFAR-10 dataset using various gradient-based sample-specific attacks.

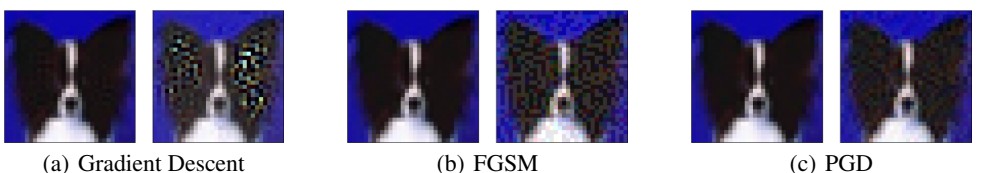

(a) Gradient Descent        (b) FGSM        (c) PGD

Figure 6: Comparison of perturbation strength sufficient to fool the network, as generated by different sample-specific gradient-based attacks. ResNet-32 is always shown on the left, whereas PR-ResNet-32 on the right.

**Fast Gradient Sign method.** For the next series of experiments, we test against the *Fast Gradient Sign Method (FGSM)* of Goodfellow et al. (2015). FGSM generates an adversarial example $\hat{\mathbf{x}}$ for an input $\mathbf{x}$ with label $y$ as follows:

$$\hat{\mathbf{x}} \leftarrow \mathbf{x} + \epsilon \operatorname{sign}(\nabla_{\mathbf{x}} J(\theta, \mathbf{x}, y)),$$

where $J(\theta, \mathbf{x}, y)$ is the cost used during training (cross-entropy in our case), and $\epsilon$ controls the magnitude of the perturbation.

**Projected Gradient Descent.** Compared to the previous, this is a much stronger adversary, often referred to as a multi-step method FGSM$^k$ (Madry et al. (2017)). Formally, given an input $x$ with label $y$, single iteration generating adversarial sample $\mathbf{x}^{t+1}$ is defined:

$$\mathbf{x}^{t+1} \leftarrow \prod_{\mathbf{x}+S} (\mathbf{x}^t + \epsilon \operatorname{sign}(\nabla_{\mathbf{x}} J(\theta, \mathbf{x}, y))),$$

where $J(\theta, \mathbf{x}, y)$ is the cost used during training (cross-entropy in our case), and $\epsilon$ controls the magnitude of the perturbation and operator $\prod$ defines the projection on a $S$-ball around $\mathbf{x}$.

**Discussion of results.** From Figure 5, we can clearly see that sample-specific gradient-based attacks have to generate much stronger perturbation in order to fool the PeerNet architectures. Some qualitative examples are shown in Figure 6. Careful reader will notice that PeerNet often cannot be fooled without generating attack observable by a human eye.

As expected, the strongest PGD attack is able to generate a perturbation for all the samples in the test set even for PeerNet. However, much stronger perturbations need to be generated. For ResNet-32, it is sufficient to set $\epsilon \geq 0.013$, while for PR-Resnet-32 we need $\epsilon \geq 0.651$, which obviously results in much stronger perturbation. This is caused by the iterative nature of PGD and also supports the fact that our method does not suffer from gradient obfuscation, as a viable perturbation can be found based on the iterative processing of network gradients for any test sample.

### 4.3.2 UNIVERSAL ADVERSARIAL PERTURBATIONS

Finally, we used the method of Moosavi-Dezfooli et al. (2016), implemented in their DeepFool toolbox to generate a non-targeted universal adversarial perturbation $\mathbf{v}$. The strength of the perturbation was bound by $||\mathbf{v}||_2 \leq \rho\mathbb{E}||\mathbf{x}||_2$, where the expectation is taken over the set of training images and the parameter $\rho$ controls the strength.

We first evaluted on MNIST dataset by LeCun (1998) and compared to LeNet-5 as baseline. The same evaluations were performed on CIFAR-10 and CIFAR-100 datasets, using ResNet-32 and ResNet-110 as baselines, respectively. We additionally compare our method on CIFAR-10 to other baselines, namely BRELU with Gaussian Additive Noise by Zantedeschi et al. (2017), and MagNet of Meng & Chen (2017). PeerNet performs significantly better than MagNet and better than BRELU for strong noise. Compared to MagNet in particular, the key advantage of our architecture is that it is fully trained end-to-end together with regularization layers instead of as a separate independent module. We argue that learned features are more discriminative and robust if trained end-to-end. Moreover, it is worth mentioning that training a separate module is very prone to gradient obfuscation problem mentioned before. Results in Figure 7 clearly show that our PeerNet variants are much more robust than the respective baselines.

We observe minor loss in accuracy at $\rho = 0$, which is a consequence of the regularization due to the averaging in feature space, and we argue that it could be mitigated by increasing the model capacity. For this purpose, the same configuration with double number of maps in PR layers (marked as v2) was trained, and results show indeed a sensible improvement that does not come at the cost of a much higher fooling rate as for the ResNet baselines.

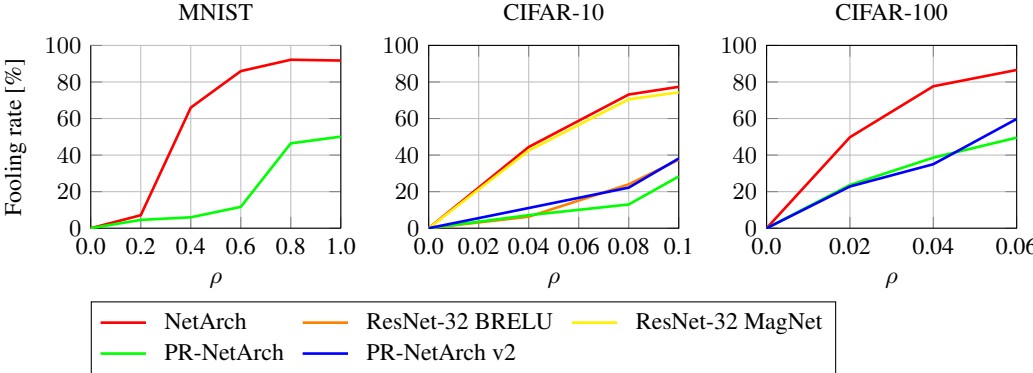

Figure 7: Fooling rate on various datasets for different levels $\rho$ of universal adversarial noise. The NetArch is Lenet-5, ResNet-32 and ResNet-110 for MNIST, CIFAR-10 and CIFAR-100 datasets respectively.

A visual comparison of perturbations generated for classical CNN and our PR-Net is depicted in Figure 4. It can be immediately appreciated that the perturbations for PR-Net have more localized structures. We argue that this is due to the attempt of fooling the KNN mechanism of the PR layers, resulting in strong noise in 'background' areas rather than in the central parts usually containing the object.

## 5 CONCLUSIONS

We introduced PeerNets, a novel family of deep networks alternating Euclidean and Graph convolutions to harness information from peer images and showed their robustness to adversarial attacks in a variety of scenarios through extensive experiments for white-box attacks in targeted and non-targeted settings. PeerNets are simple to use and can be added to any baseline model with minimal changes and are able to deliver remarkably lower fooling rates with negligible loss in performance. Interestingly, the amount of noise required to fool PeerNets is much higher and results in the generation of new images where the noise has clear structure and is significantly more perceivable to the human eye. In future work, we plan to provide a theoretical analysis of the method and scale it to ImageNet-like benchmarks.

ACKNOWLEDGMENTS

We gratefully acknowledge the generous support by ERC Consolidator Grant No. 724228 (LEMAN), multiple Google Research Faculty awards and Nvidia equipment grants. MB is also partially supported by the Royal Society Wolfson Research Merit award and Rudolf Diesel industrial fellowship at TU Munich. LG is supported by NSF grant DMS-1546206, a Vannevar Bush Faculty Fellowship, and a gift from Amazon Web Services

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

## A  TRAINING HYPER-PARAMETERS

The training hyper-parameters used in our experiments are summarized in Table 1.

Table 1: Optimization parameters for different architectures and datasets. Learning rate is decreased at epochs 100, 175, and 250 with a step factor of $10^{-1}$.

| Model | Optimizer | Epochs | Batch | Momentum | LR | $L_2$ reg. | LR decay |
|---|---|---|---|---|---|---|---|
| LeNet-5 | Adam | 100 | 128 | — | $10^{-3}$ | $10^{-4}$ | — |
| PR-LeNet-5 | Adam | 100 | 32 | — | $10^{-3}$ | $10^{-4}$ | — |
| ResNet-32 | Momentum | 350 | 128 | 0.9 | $10^{-1}$ | $10^{-3}$ | step |
| PR-ResNet-32 | Momentum | 350 | 64 | 0.9 | $10^{-2}$ | $10^{-3}$ | step |
| ResNet-110 | Momentum | 350 | 128 | 0.9 | $10^{-1}$ | $2 \times 10^{-3}$ | step |
| PR-ResNet-110 | Momentum | 350 | 64 | 0.9 | $10^{-2}$ | $2 \times 10^{-3}$ | step |

It worths noticing that due to the limited GPU memory, we typically had to decrease the batch size for the PR network architectures.

## B  SAMPLE-SPECIFIC GRADIENT-BASED PERTURBATIONS

| Attack | ResNet-32 | | | PR-ResNet-32 | | |
|---|---|---|---|---|---|---|
| | Fool. rate [%] | $\mathbb{E}\|\mathbf{v}\|_2$ | $\mathbb{E}\|\mathbf{v}\|_\infty$ | Fool. rate [%] | $\mathbb{E}\|\mathbf{v}\|_2$ | $\mathbb{E}\|\mathbf{v}\|_\infty$ |
| Grad.Desc. | 74.90 | 118.39 | 15.69 | 29.80 | 238.72 | 39.39 |
| FGSM | 98.97 | 213.22 | 3.93 | 93.04 | 627.41 | 11.90 |
| PGD | 100 | 30.06 | 0.62 | 100 | 400.32 | 16.25 |

Table 2: Comparison of PeerNet and classical CNN on the CIFAR-10 dataset using various gradient-based sample-specific attacks.

## C  UNIVERSAL ADVERSARIAL PERTURBATIONS

Results for universal adversarial perturbations attacks using dataset MNIST are in Table 3, for CIFAR-10 in Table 4 and CIFAR-100 in Table 5.

Table 3: Performance and fooling rates on the MNIST dataset for different levels $\rho$ of universal adversarial noise.

| Method | Original Accuracy | Accuracy / Fooling Rate | | | | |
|---|---|---|---|---|---|---|
| | | $\rho = 0.2$ | $\rho = 0.4$ | $\rho = 0.6$ | $\rho = 0.8$ | $\rho = 1.0$ |
| LeNet-5 | 98.6% | 92.7% / 7.1% | 33.9% / 66.0% | 14.1% / 85.9% | 7.9% / 92.2% | 8.2% / 91.7% |
| PR-LeNet-5 | 98.2% | 94.8% / 4.6% | 93.3% / 6.0% | 87.7% / 11.7% | 53.2% / 46.4% | 50.1% / 50.1% |

Table 4: Performance and fooling rates on the CIFAR-10 dataset. ResNet-32 v2 and PR-ResNet-32 v2 have double the amount of feature maps after the last two convolutional blocks, meaning instead of (16, 16, 32, 64), it has (16, 16, 64, 128).

| Method | Graph size | MC runs | Acc. orig [%] | Acc pert. [%] / Fool rate [%] | | |
|---|---|---|---|---|---|---|
| | | | | $\rho = 0.04$ | $\rho = 0.08$ | $\rho = 0.10$ |
| ResNet-32 | N/A | N/A | 92.73 | 55.27 / 44.42 | 26.84 / 73.14 | 22.74 / 77.34 |
| ResNet-32 v2 | N/A | N/A | 94.17 | 44.51 / 55.32 | 16.65 / 83.40 | 12.58 / 87.58 |
| BRELU + GDA | N/A | N/A | 87.44 | 87.01 / 6.30 | 74.57 / 23.99 | 61.91 / 37.48 |
| MagNet | N/A | N/A | 92.70 | 57.09 / 42.50 | 29.48 / 70.50 | 25.61 / 74.35 |
| PR-ResNet-32 | 50 | 1 | 88.18 | 87.27 / 7.98 | 82.43 / 14.08 | 69.33 / 28.80 |
| PR-ResNet-32 | 50 | 10 | 89.30 | 87.27 / 7.13 | 83.32 / 12.99 | 70.01 / 28.31 |
| PR-ResNet-32 | 100 | 5 | 89.19 | 87.33 / 7.43 | 83.37 / 13.20 | 70.11 / 28.19 |
| PR-ResNet-32 v2 | 50 | 10 | 90.72 | 85.26 / 11.05 | 75.46 / 22.20 | 60.75 / 38.14 |
| PR-ResNet-32 v2 | 100 | 5 | 90.65 | 85.35 / 11.25 | 75.94 / 21.82 | 61.10 / 37.77 |

Table 5: Performance and fooling rates on the CIFAR-100 dataset. PR-ResNet-110 v2 has double the amount of feature maps after the last two convolutional blocks, meaning instead of (16, 16, 32, 64), it has (16, 16, 64, 128).

| Method | Graph size | MC runs | Acc. orig [%] | Acc pert. [%] / Fool rate [%] | | |
|---|---|---|---|---|---|---|
| | | | | $\rho = 0.02$ | $\rho = 0.04$ | $\rho = 0.06$ |
| ResNet-110 | N/A | N/A | 71.63 | 45.49 / 49.78 | 20.99 / 77.64 | 12.74 / 86.56 |
| PR-ResNet-110 | 500 | 5 | 66.40 | 61.47 / 23.65 | 52.61 / 38.59 | 44.64 / 49.54 |
| PR-ResNet-110 v2 | 500 | 5 | 70.66 | 63.71 / 22.84 | 56.40 / 35.01 | 36.74 / 59.76 |

## D ADVERSARIAL TRAINING

We have compared our approach to adversarial training method using the code[1] provided by Madry et al. (2017). The ResNet-32 baseline model provided by TensorFlow (the same we use as CNN baseline in our paper) was trained using the script provided in the repository mentioned above. We have used two training configurations producing two baseline models:

1. *ResNet-32 A* - the default hyperparameters provided by the repository
2. *ResNet-32 B* - the same hyperparameters as in our paper

PeerNet model was trained traditionally using SGD with momentum without adversarial training.

Selection and configuration of the attack was left as provided by the repository mentioned above.

Results shown in the Table 6 show superiority of PeerNet on this benchmark. Moreover, PR-ResNet was trained without considering any specific attacks and still outperforms ResNet-32, which was adversarially trained using this specific attack, by margin of 20%.

Table 6: Comparison of PeerNets to adversarially trained version of ResNet-32 baseline on CIFAR-10 dataset.

| Method | Acc. orig. [%] | Acc. pert. [%] |
|---|---|---|
| ResNet-32 A | 78.86 | 45.47 |
| ResNet-32 B | 75.59 | 42.53 |
| PR-ResNet | 77.44 | 64.76 |

## E BLACK-BOX ATTACKS

For completeness, additional comparison of our method to some of the strong black box attacks has been carried out using repository[2] by Bhagoji et al. (2017).

The evaluation has been done on small subset of 100 test samples. We have used untargeted both single-step and iterative query black-box attack computed using finite difference method, compared to CW-loss based white-box attack. The results are listed in the Table 7.

Table 7: Evaluation of PeerNets on finite-difference based query black-box attacks using CIFAR-10 dataset.

| Method | Attack type | $\epsilon$ | Fooling rate [%] | |
|---|---|---|---|---|
| | | | White-box | Black-box |
| ResNet-32 | single step | 8.0 | 82.00 | 82.00 |
| PR-ResNet | single step | 8.0 | 34.00 | 14.00 |
| ResNet-32 | iterative | 0.5 | 41.00 | 41.00 |
| ResNet-32 | iterative | 8.0 | 100.00 | 100.00 |
| PR-ResNet | iterative | 8.0 | 45.00 | 28.00 |
| PR-ResNet | iterative | 12.0 | 63.00 | 37.00 |

The results above show that our method is efficient of black-box attacks as well. Moreover, it verifies that our method does not cause gradient masking. Athalye et al. (2018) mentions that iterative attacks should be stronger than single step, which in the results above holds. Also, it states that black-box attacks should be strict subset of white-box attacks, which is confirmed as well.

---

[1]https://github.com/MadryLab/cifar10_challenge
[2]https://github.com/sunblaze-ucb/blackbox-attacks

# F  UNIVERSAL ADVERSARIAL PERTURBATIONS FOR CIFAR-10 VISUALLY

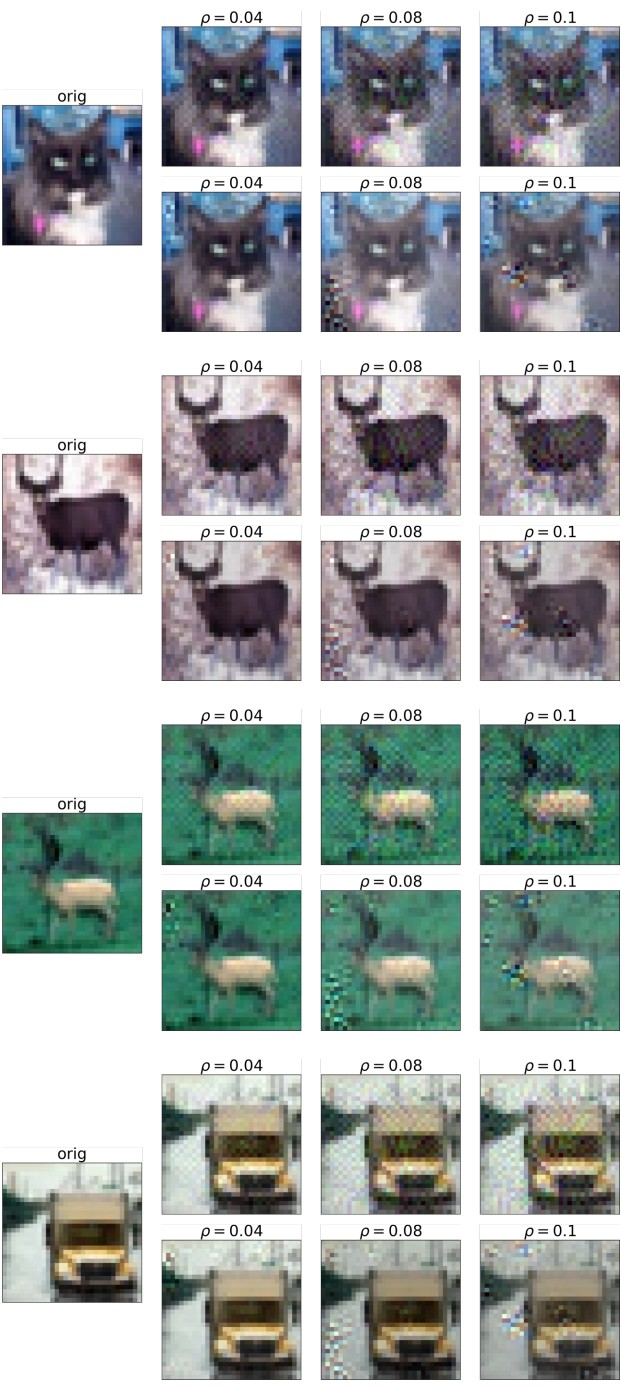

Figure 8: Examples of applying universal perturbations to different images from the CIFAR-10 test set. For each sample, the leftmost is the original image, then adversarial examples for ResNet-32 are shown in the top row, whereas for PR-ResNet-32 in the bottom row.

## G NON-TARGETED SAMPLE-SPECIFIC PERTURBATIONS FOR CIFAR-10 VISUALLY

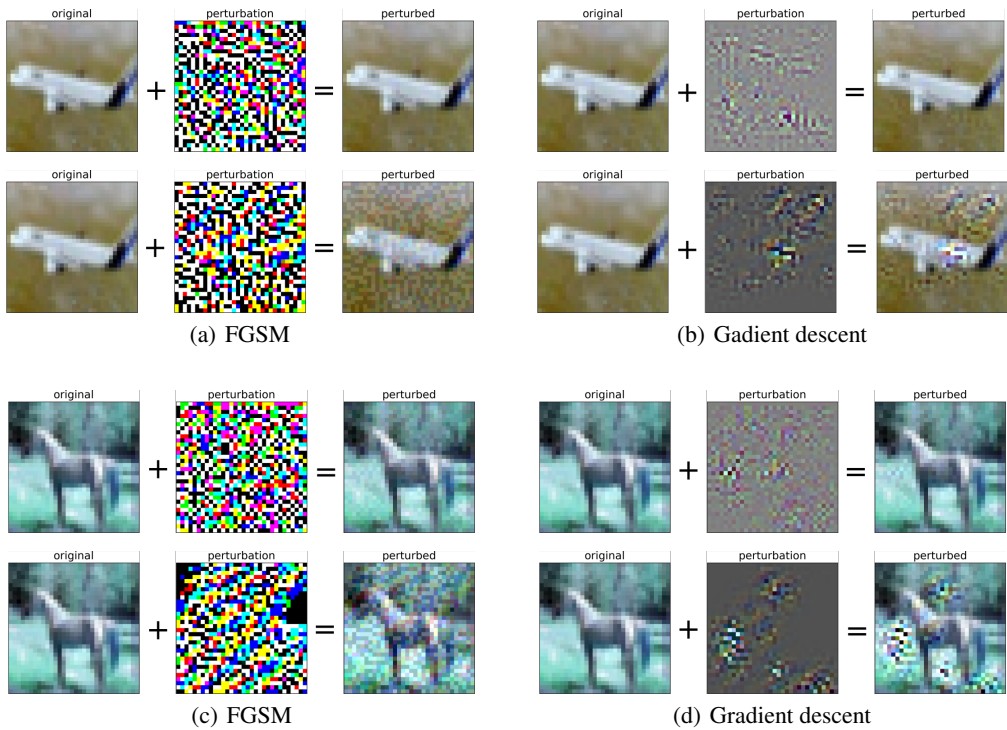

Figure 9: Examples of sample-specific gradient-based perturbations for different images from the CIFAR-10 test set using FGSM((a), (c)) and gradient descent((b), (d)). For each sample, results for ResNet-32 are shown in the top row, whereas for PR-ResNet-32 in the bottom row. In order to successfully generate adversarial samples for PR-Nets, the magnitude of the perturbation has to be much higher.

