# OpenReview forum: "PeerNets: Exploiting Peer Wisdom Against Adversarial Attacks"
_ICLR.cc/2019/Conference_

### Official Review · AnonReviewer2 · 2018-10-28
**Analysis and experimental comparisons are lacking**

**Rating:** 6
**Confidence:** 5

**Review:**

After reading the authors' response, I'm revising my score upwards from 5 to 6.

The authors propose a defense against adversarial examples, that is inspired by "non local means filtering". The underlying assumption seems to be that, at feature level, adversarial examples manifest as IID noise in feature maps, which can be "filtered away" by using features from other images. While this assumption seems plausible,  no analysis has been done to verify it in a systematic way. Some examples of verifying this are:

1. How does varying the number of nearest neighbors change the network behavior?
2. At test time, a fixed number of images are used for denoising - how does the choice of these images change accuracy or adversarial robustness?
3. Does just simple filtering of the feature map, say, by local averaging, perform equally well?
4. When do things start to break down? I imagine randomly replacing feature map values (i.e. with very poor nearest neighbors) will cause robustness and accuracy to go down - was this tested?

Based on the paper of Athalye et. al., really the only method worth comparing to for adversarial defense, is adversarial training. It is hard to judge absolute adversarial robustness performance without a baseline of adversarial training.

---

> ### Author Response · Authors · 2018-11-20
> **Requested analysis and comparison**
>
> Thank you for the valuable remarks.
>
> We have tested most of the concerns in points 1. - 4. during our experiments. We however could not provide full-extent analysis due to the limited length of the paper. Let us respond to each of the points separately below.
>
> 1. How does varying the number of nearest neighbors change the network behavior?
> ---
> We observe that using small k (~5) doesn't always provide enough information to perform the denoising and the network is therefore less robust against adversarial examples.
> On the other hand, having k too high (~20) yields too much regularization and the network original performance decreases more significantly.
> In our experiments, we have found k=10 to be a reasonable compromise.
>
> 2. At test time, a fixed number of images are used for denoising - how does the choice of these images change accuracy or adversarial robustness?
> ---
> We refer the reviewer to the Section 3 and Section 4.1. of our paper where this is addressed in detail.
>
> 3. Does just simple filtering of the feature map, say, by local averaging, perform equally well?
> ---
> It does not. We have tried simple smoothing of the feature maps and it not only does not make the network robust against adversarial attacks, but also regularizes the original network too much which results in significant loss in classification accuracy. Moreover, local averaging uses the information from the corrupted image itself to filter the feature map, which could even further amplify the noise.
>
> 4. When do things start to break down? I imagine randomly replacing feature map values (i.e. with very poor nearest neighbors) will cause robustness and accuracy to go down - was this tested?
> ---
> This is of-course true. Obviously, selecting very poor nearest neighbors will definitely break the method as the newly created feature map will not express the original information anymore. In our paper, we even reason that the adversary often tries to fool the KNN algorithm directly, as we mention at the end of Section 4.3.2.
> Moreover, we believe that our results show when do things start to "break down". We explicitly mention that an unbounded attack will always fool the network. Also, our figures in the main text and tables in the supplementary material show that with increasing magnitude of the perturbation, things start to "break down".
>
> 5. Based on the paper of Athalye et. al., really the only method worth comparing to for adversarial defense, is adversarial training. It is hard to judge absolute adversarial robustness performance without a baseline of adversarial training.
> ---
> We provide an evaluation below as well as add an additional section with the results in the supplementary material.
>
> We have compared our approach to adversarial training method using the code provided by Madry etal. https://github.com/MadryLab/cifar10_challenge.
> The ResNet-32 baseline model provided in Tensorflow repository (the same we use as CNN baseline in our paper) was trained using the script provided in the cifar10_challenge repository above.
> We have used two training configurations producing two baseline models1 - the default one provided by the repository (ResNet-32 CNN A) and then the same one as in our paper (ResNet-32 CNN B).
> PeerNet was trained traditionally without adversarial training.
> The attack was left as defined by the repository by Madry etal.
>
> ResNet-32 CNN A: original_acc = 78.86% | adversarial_acc = 45.47%
> ResNet-32 CNN B: original_acc = 75.59% | adversarial_acc = 42.53%
> PeerNet:         original_acc = 77.44% | adversarial_acc = 64.76%
>
> Results show superiority of PeerNet on this benchmark. PeerNet was trained without considering any specific attacks and still outperforms ResNet-32 CNN, which was adversarially trained using this specific attack, by margin of 20%.

---

### Official Review · AnonReviewer1 · 2018-11-02
**interesting work introducing graph neural nets as regularization, with practical limitations**

**Rating:** 7
**Confidence:** 4

**Review:**

The paper presents a an interesting novel approach to train neural networks with so called peer regularization which aims to provide robustness to adversarial attacks. The idea is to add a graph neural network to a spatial CNN. A graph is defined over similar training samples which are found using a Monte Carlo approximation.

The regularization using graphs reminds me of recent work at ICML on semi-supervised learning (Kamnitsas et al. (2018) Semi-supervised learning via compact latent space clustering) which is using a graph to approximate cluster density which acts as a regularizer for training on labelled data.

The main problem I see with these approaches is that they rely on sufficiently large batch sizes which could be (currently) problematic for many real-world applications. Memory and computation limitations are mentioned, but not sufficently discussed. It would be good to add further details on practical limitations.

Experiments are limited to benchmark data using MNIST, CIFAR-10, CIFAR-100. Comprehensive evaluation has been carried out with insightful experiments and good comparison to state-of-the-art. Both white- and black-box adversarial attacks are explored with promising results for the proposed approach.

However, it is difficult to draw conclusions for real-world problems of larger scale. The authors state that proposed framework can be added to any baseline model, but miss to clearly mention the limitations. It is stated that future work will aim at scaling PeerNets to benchmarks like ImageNet, but it is unclear how this could be done. Is there any hope this could be applied to problems like 3D imaging data or videos?

---

> ### Author Response · Authors · 2018-11-20
> **Scalability of our method**
>
> Thank you for the insightful comments.
>
> 1. It is stated that future work will aim at scaling PeerNets to benchmarks like ImageNet, but it is unclear how this could be done. Is there any hope this could be applied to problems like 3D imaging data or videos?
> ---
> Regarding the concerns on method scalability. The current bottleneck of our approach is processing of all feature maps pixel-wise. We see potential of scaling our approach by operating on superpixels, or NxN patches, instead of processing all pixels individually.

---

> > ### Comment · AnonReviewer1 · 2018-11-29
> > **please elaborate on all concerns**
> >
> > I am a bit disappointed by the author's response which only picks up one of my points raised in the review.
> >
> > 1) I would have liked to see a response regarding my point on batch sizes? Is that a problem at all in your approach?
> >
> > 2) What conclusions can we draw from MNIST and CIFAR for real-world applications?

---

> > > ### Author Response · Authors · 2018-11-29
> > > **Answering other concerns**
> > >
> > > We apologize to the reviewer for our brief response. We have tried to address what we believed to be the most prominent concern, thus inadvertently overseeing some other important comments.
> > >
> > > Please find answers to the additional concerns below.
> > >
> > > 1. I would have liked to see a response regarding my point on batch sizes? Is that a problem at all in your approach.
> > > ---
> > > As we mention in our paper, we used somewhat smaller batch sizes than traditional CNNs, primarily due to the current memory requirements and non-optimized implementation of our code.
> > > The attention weights do not depend on number of peers in the graph and we can therefore change batch size dynamically as we want. We have tried batch sizes as small as 8 samples without seeing dramatic decrease in performance.
> > > During test time, we have tried different number of peers in the graph - 10,50,100,200,500. We did not observe any significant changes in performance, please refer to some additional results in Table 4 of Appendix C.
> > >
> > > 2. What conclusions can we draw from MNIST and CIFAR for real-world applications?
> > > ---
> > > We believe that key novelty of our work is a deep-learning network architecture where we can interleave graph layers with convolutional layers and learn everything end-to-end. To our knowledge, this is the first such architecture, and it has many potential use-cases, e.g. image inpainting, few-shot learning, adversarial defense, etc. We have chosen to showcase our architecture on a very hot topic of adversarial attack defense.
> > > It should be noted that a vast majority of the defense mechanisms described in the literature or available online do not show any results on ImageNet but rather smaller benchmarks such as CIFAR. We have therefore focused on such datasets, in order to provide if not direct comparison, at least a ballpark compared to the current state-of-the-art.
> > > We agree with the reviewer that it is hard to draw conclusions regarding real-world applications of our approach from the datasets we have tested on. We however firmly believe that extending our approach with ideas mentioned in our comment on scalability should make our method a good fit for real-world scenarios.

---

### Public Comment · (anonymous) · 2018-10-26
**Neither MagNet nor BRELU are effective**

For what it's worth, both of these defenses you compare against are ineffective ( https://arxiv.org/abs/1711.08478 ).

---

> ### Author Response · Authors · 2018-11-05
> **Baseline comparisons**
>
> Thanks for suggesting the reference, which we will include in the revision. Considering the paper by Athalye et al. (https://arxiv.org/abs/1802.00420), most of the recently proposed defenses are ineffective. We actually refer to this fact in Section 4.3.2, where the 2nd paragraph points out that the way we train our model should make it more robust compared to the baselines, which, by the nature of their use, are prone to the problem of obfuscated gradients.
>
> We however needed to select some baselines to compare to, and chose the methods that are, in some sense, similar to our approach.

---

### Public Comment · (anonymous) · 2018-10-26
**Comparison with Madry et al. 2017**

Can you compare with adversarial training? Also, did you vary the PGD iterations and verify that the accuracy does not change?

Further, have you considered the CW attack, gradient-free attacks and adaptive versions?

---

> ### Author Response · Authors · 2018-11-05
> **Additional comparison**
>
> Thank you for your comments.
>
> Can you compare with adversarial training?
> ---
>
> 1. We tried ResNet-32 with adversarial training. The resulting combination is not efficient, as also reported by Dezfooli et al.
>
> 2. We used the default parameters suggested by foolbox. During PGD iterations, we have observed that the update in perturbation strength (increase or decrease) is diminishing into orders of 1-e3 over the iterations and we therefore did not expect that increasing the  number of iterations would give us any noticeable improvement.
>
> Further, have you considered the CW attack, gradient-free attacks and adaptive versions?
> ---
>
> 3. In our work, we mainly focused on gradient-based white box attacks and compared primarily against universal adversarial perturbations by Dezfooli etal. and also against some of the classic gradient-based attacks (gradient descent, FGSM, PGD). We have not tried gradient-free attacks.
>
> Following your suggestion, we evaluated out method on CW L2 attack, which is newly implemented in foolbox. To provide a fair comparison, we used the same settings for PeerNets and CNN baseline. The results are reported below:
>
> CNN baseline: L2 error = 49.86 | Linf error = 4.327 | fooling rate = 100%
> PeerNets: L2 error = 365.59 | Linf error = 27.43 | fooling rate = 93.76%
>
> This clearly shows that PeerNets are more robust to CW L2 attack (the perturbation required to achieve a similar fooling rate on PeerNet has to be nearly an order of magnitude stronger).

---

### Public Comment · (anonymous) · 2018-11-14
**The defense appears to be causing gradient masking**

It appears this paper is causing gradient masking. Looking at Figure 5, FGSM is a more effective attack than PGD at eps=0.1, which is highly suspicious and is listed as one of the tests for identifying gradient masking in Athalye et al. 2018.

The authors may wish to try black-box attacks (e.g., SPSA by Uesato et al at ICML'18) to see if this is happening.

Further, this paper claims ~15% robustness at eps=0.1 (25/255) on CIFAR-10, and a non-zero robustness at eps=0.2 (50/255). No prior work has been able to achieve greater than ~10% accuracy at eps=0.06 (16/255), see Madry et al. 2018.

---

> ### Author Response · Authors · 2018-11-20
> **Additional evaluation suggests the defense does not cause gradient masking**
>
> Thank you for your comments.
>
> Based on your concerns, we below provide additional evaluation to show that our approach does not cause gradient masking. We also add the new results to the supplementary material of our paper.
>
> 1. It appears this paper is causing gradient masking. Looking at Figure 5, FGSM is a more effective attack than PGD at eps=0.1, which is highly suspicious and is listed as one of the tests for identifying gradient masking in Athalye et al. 2018.
> ---
> Regarding your concern to the Figure 5, where FGSM seems more effective than PGD at eps=0.1. We attribute this to the selection of the hyperparameters of the FGSM and PGD attack. The number of iterations per each value of epsilon for the PGD attack was set to 40. We have tried to increase this parameters to 100 iterations and evaluate on a small subset of the test set. This causes the PGD to find a slightly better perturbations in many cases, and slightly improves the attack on PeerNet. The overall comparison PeerNet vs. CNN baseline however does not change significantly, and we therefore choose to leave the default configuration of 40 iterations, as it is much more computationally feasible for the full test set of 10000 samples.
>
> 2. The authors may wish to try black-box attacks (e.g., SPSA by Uesato et al at ICML'18) to see if this is happening.
> ---
> Further, to support our claim, we have taken this repository implementing black box attacks, including SPSA mentioned by the reviewer https://github.com/sunblaze-ucb/blackbox-attacks .
> We have used their code for cifar-10 dataset and performed evaluation on a subset of 100 test samples. Using untargetted both single-step and iterative query black-box attack computed using finite difference method, compared to CW-loss based white-box attack.
> The code reports "fraction of targtets achieved" for both white-box (whitebox_succ) and black-box (blackbox_succ) attacks, which is the percentage of samples that the attack was successful on.
> We have left the default attack configuration from the repository, using epsilon = 8.0.
> Single step:
> ResNet-32 CNN (eps=8.0): whitebox_succ = 82% | blackbox_succ = 82%
> PeerNet (eps=8.0):       whitebox_succ = 34% | blackbox_succ = 14%
>
> For the iterative attack, we have tried different values of epsilon for CNN and PeerNet to allow fair comparison of the attacks
> Iterative:
> ResNet-32 CNN (eps=0.5): whitebox_succ = 41%  | blackbox_succ = 41%
> ResNet-32 CNN (eps=8.0): whitebox_succ = 100% | blackbox_succ = 100%
> PeerNet (eps=8.0):       whitebox_succ = 45%  | blackbox_succ = 28%
> PeerNet (eps=12.0):      whitebox_succ = 63%  | blackbox_succ = 37%
>
> From the results above, we verify that our method does not cause gradient masking. Athalye et al. 2018 mentions that iterative attacks should be stronger than single step, which in the results above holds. Moreover, it states that black-box attacks should be strict subset of white-box attacks, which is confirmed as well.
>
> 3. Further, this paper claims ~15% robustness at eps=0.1 (25/255) on CIFAR-10, and a non-zero robustness at eps=0.2 (50/255). No prior work has been able to achieve greater than ~10% accuracy at eps=0.06 (16/255), see Madry et al. 2018.
> ---
> We are not aware of any proof that would set the theoretical bound of ~10% accuracy at eps=0.06 on CIFAR-10. We attribute the significant gap to the superiority of our approach. The significant margin can be potentially partially reduced by tuning hyperparameters of the attack methods. Nevertheless, from our observations, PeerNet will still stay very robust.

---

### Meta-Review · Area_Chair1 · 2018-12-17
**Accept**

**Confidence:** 5
**Recommendation:** Accept (Poster)

**Metareview:**

The paper presents a novel with compelling experiments. Good paper, accept.